# The Wear on Roller Press Rollers Made of 20Cr4/1.7027 Steel under Conditions of Copper Concentrate Briquetting

**DOI:** 10.3390/ma13245782

**Published:** 2020-12-17

**Authors:** Michał Bembenek, Janusz Krawczyk, Krzysztof Pańcikiewicz

**Affiliations:** 1Faculty of Mechanical Engineering and Robotics, AGH University of Science and Technology, A. Mickiewicza 30, 30-059 Kraków, Poland; 2Faculty of Metals Engineering and Industrial Computer Science, AGH University of Science and Technology, A. Mickiewicza 30, 30-059 Kraków, Poland; jkrawcz@agh.edu.pl (J.K.); krzysztof.pancikiewicz@agh.edu.pl (K.P.)

**Keywords:** roller press rollers, wear of rollers, 20Cr4 microstructure, steel surface hardening, high pressure wear, briquetting

## Abstract

This paper defines the wear process of rollers made of 20Cr4. Rollers with a diameter of 1000 mm were installed in a roller press used for the production of drop-shaped briquettes and the copper concentrate was briquetted for 1100 h. Three-dimensional (3D) geometry analysis, metallographic analysis, macroscopy, scanning electron microscopy, as well as hardness measurements were performed. It was observed that the working surface was non-uniformly worn. The smallest wear affects the molding cavities situated on the outermost edges of the ring. The wear increases as the center of the ring is approximated, and it reaches its maximum at the middle of the ring. The molding cavities also wear asymmetrically. For the shape considered in this study, the lower part of a cavity is subject to a higher wear rate. We found that the material of the working ring was carburized, but its hardness was significantly lower than required. The roller ring microstructure changes depended on the distance from the cavity’s face. An investigation of the wear mechanisms showed different types of abrasive wear, corrosive processes, and plastic deformation. The exact type and course of wear were described, depending on the location on the working surface.

## 1. Introduction

There are many processes where input material is required to come in the form of large pieces instead of a loose form [1,2,3]. Therefore, agglomeration is a key process that is presently employed in power [4,5,6,7,8], heavy [9,10,11,12], chemical [13,14] and pharmaceutical industries [15,16,17]. The method widely used to consolidate loose materials is pressurized agglomeration [1,2,3,4]. It consists of putting the material under pressure, causing the grains to be closer to one another, and forming diverse bonds to make the material stronger [18]. A very important element of a properly conducted agglomeration is effective preparation of the material [11,13,19] and its dosing [17,20,21]. The pressurized agglomeration process produces compact shapes with relatively low porosity and specific mechanical strength [6,12,18]. These shapes can be sized several thousand times larger as compared with the input material grains. Industries have willingly used roller presses rather than other briquetting machines, for example, screw or punch briquette machines [3,9,11,13,14,15,16]. Machines of this type demonstrate constant operability with one of their major advantages being a relatively low demand for energy [3,8,9].

The compaction unit is a particularly important unit among the systems for pressurized agglomeration [8,15,21], which is where the production of the agglomerate takes place. It is also the weakest link in a briquetting machine’s structure. A standard compaction system consists of molding elements and a feeder for supplying properly preconditioned fine-grained material on a continuous basis. Presses with two working rollers are the ones that are most frequently used; however, mono-roller or multi-roller solutions are also used. In briquetting machines, the rollers are parallel to one another and rotate in opposite directions with the same rotational speed rate. The working part of the rollers is most often made in the form of monolithic rings; segments installed on special fittings are less often used [3]. In traditional roller presses, the cavities on working surfaces of rollers are deployed to be their mutual mirror images on both rollers [22]. The compaction unit of this type is referred to, in the literary sources, as symmetrical [3,23]. The unit pressure in the fine-grained material consolidation process often reaches several hundred MPa and is a parameter which results, among other things, from the properties of the material being consolidated and the compaction system geometry [8,15,24]. Its value changes depending on the location on the molding surface [25]. The unit pressure changeability results from the occurrence of different zones of compaction in the cavity where the briquette is being molded [24], the various compaction phases during the rotation of molding rings [25], the size and location of dead fields between particular molding cavities, and the non-homogeneity of the material being consolidated. The cavities arranged on the cylindrical surface form a specific pattern. There are thresholds and nodes among them. The occurrence of thresholds and nodes among the cavities results in rhythmic impact loads acting on the rollers during the operation. These stresses reach their peak when the opposite thresholds of both rollers, in the roller press, come into contact [15,16]. This negative phenomenon is limited by the consecutive rows of cavities being alternately arranged on the roller surface [21,26]. Such arrangement makes it possible to minimize the areas of dead fields which form thresholds and nodes. In briquettes molded using a classic symmetrical compaction system, a parting line appears [23]. This is an unfavorable property related to the formation of a briquette which is fused from the material situated in two opposite cavities. Such briquettes tend to split in half along the parting lines after they leave molding cavities [2]. In the molding cavity, unit pressures with varying values occur. The lowest unit pressure value occurs in the lower area of the cavity (Figure 1). It can result in a non-homogeneous distribution of density in the briquette and a loss of its mechanical strength [24]. To create a homogeneous unit pressure homogeneous, classic cushion-shaped molding cavities can be shallowed in the lower zone of the molding cavity [1]. Such a solution also serves to extend the period of operation of the molding rings and has a favorable effect on the limitation of stratification of the briquette along the parting line. Briquettes in shallowed compaction molds are shaped like a water drop.

During briquetting of fine-grained materials, the working surface gets intensely worn out [27,28,29]. It has been assumed that they are primarily subjected to abrasive, fatigue, and corrosion-related wear. On the working surface of rollers, the thresholds and nodes are referred to as the so-called dead fields which are exposed to an intense wear process since maximum values of the friction force and high unit pressure occur in these locations. These are the areas where the working rings become subject to the wear process [30]. The wear leads to plastic deformation, damage, and abrasive wear of thresholds and nodes. The shape and dimensions of the molding cavities also change. The molding cavities are worn in a repeatable way, mainly in their lower part. This is possibly related to the consolidated material flowing out of the consolidation zone during the operation in the final phase of the process (the interacting semi-molds in both rollers are open). The briquettes expand and try to escape from the molding cavity. This is also related to the elastic expansion of briquettes [23], which causes them to move against the working surface and is conducive to the abrasive wear. The wear of the working surface of the molding rings results from a reduction in the thickness of walls separating particular cavities [30]. The unit pressure exerted in the molding cavities is not identical along the entire forming length of the roller and reaches its peak in the middle of the roller because the flow of the material being consolidated is most intense at this point. The slowest flow of material occurs at the walls of the feeder side sealings. This results from the friction force acting between the material and the feeder walls, which causes the flow to brake in these zones [31]. On the basis of this, it is possible to explain why the wear of the rings increases away from the feeder walls towards their center [30]. The outermost cavities show the slowest wear rate, which results in the rings showing their highest degree of wear in their central part. This causes the diameter of the rings in their central part to decrease [30,31].

The wear of the molding surfaces is related to the deterioration of the mechanical properties of briquettes. The volume of the forming cavities increases with the wear of the molding surfaces. Increasing the volume of briquettes reduces the unit pressure, which in extreme cases may be insufficient to properly merge the briquette or can cause the material to stay in the worn molding cavities. This results in an increase in loads on the compaction systems and the drive system of the press as well as a decrease in its output capacity [22,32]. The wear of the molding surface can also lead to an increase in the gap between the rolls, which also results in a reduction in the pressures exerted in the molding cavity. Our experience shows that the wear of rollers is greatly influenced by the properties of the briquetting material, especially the external friction coefficient, hardness, and abrasive properties. The actions aiming at accurate identification of wear mechanisms and also the intensity of these phenomena over time need to be accurately understood and accurately verified using real objects.

The annual global production of copper is about 20 m tons of copper, of which 590 thousand tons are produced in Poland and, of these, over 120 thousand tons are currently produced by means of the shaft furnace copper concentrate smelting technology [33]. Once the originally extracted copper ore is ground and enriched, it becomes fine-grained concentrate. It is fed into shaft furnaces used to smelt copper matte after it has been transformed into lumps [34] by briquetting with the use of liquid sulphite lye and a binder [33]. However, current tests in the field of wear analysis of molding rings have only been carried out in a narrow scope [30,31]. The available tests for carburizing steels with similar composition or properties have mainly concerned laboratory tests at appropriate stands [35,36,37,38,39,40,41,42]. Since the copper smelting industry is an important and strategic sector of the heavy industry, it has been decided to carry out an accurate wear analysis of the molding rings which are used in the industrial environment for the briquetting of copper ore concentrate.

The purpose of this work was to analyze the wear of molding rollers performed by carburized 20Cr4 steel. This goal was achieved by 3D geometry analysis, macroscopic observations, optical emission spectroscopy, metallographic microscopic examinations with light and scanning electron microscopy, and hardness measurements.

## 2. Materials and Methods

The wear assessment tests were carried out covering a sector of the molding ring working in a symmetrical compaction system with a rated diameter of 1000 mm and a working width of 580 mm. The molding ring was installed in a four-roller roller press (a press with two compaction systems operating in parallel) with a capacity of 70 Mg/h, drive power of 2 × 75 kW, total pressure force of drums of 8 MN, and was working with a rotational speed of 4.8 RPM (Figure 2).

The working surface of each ring contains 344 water drop shaped cavities arranged in eight rows with 34 cavities each, each cavity having a capacity of 60 cm^3^ (120 cm^3^ in total for each briquette) (Figure 3). The ring was made of 20Cr4 steel (1.7027) and its surface was carburized to a depth of 2 mm. Then, the ring underwent heat treatment (hardening and low tempering). According to production technology, the molding surface hardness should amount to 700 HV. The ring chemical composition, identified with the Foundry-Master (WAS) optical emission spectrometer (Hitachi, Tokyo, Japan), is presented in Table 1. The values of alloy elements fall within a standard range established for 20Cr4 steel [44].

The ring worked for about 1100 h in an industrial environment. The material subjected to briquetting was copper ore concentrate with sulphite lye with a moisture content of about 4.0–4.5%. Each of the 344 molding cavities was estimated to have produced about 330 thousand briquettes. Laboratory tests were carried out in a roller press, with rings of 450 mm diameter with cushion-shaped molding cavities with a capacity of 13 cm^3^, and showed that the pressure exerted on the bottom of the molding cavity, during the briquetting of the copper ore concentrate with sulphite lye with moisture contained within a favorable range, was usually 25–30 MPa. Since the scale of the geometrical similarity, i.e., Equation (1), of the diameter and capacity of both presses, which is shown by Equations (2) and (3), as well as taking into account a small shallowing of the briquettes with a capacity of 120 cm^3^, it can be concluded that the pressures exerted in the molding cavities of the press with a 1000 mm diameter of rollers were similar as follows:(1)$$\frac{D_{1}}{D_{2}}\;≅\;\sqrt[{3}]{\frac{V_{1}}{V_{2}}}$$
(2)$$\frac{D_{1}}{D_{2}}\;=\;\frac{1000\;\mathrm{mm}}{450\;\mathrm{mm}}=2.22$$
(3)$$\sqrt[{3}]{\frac{V_{1}}{V_{2}}}=\sqrt[{3}]{\frac{120\;{\mathrm{cm}}^{3}}{13\;{\mathrm{cm}}^{3}}}=2.10$$
where D_1_ and D_2_ are roller diameters (mm) and V_1_ and V_2_ are briquette volumes (cm^3^).

This was confirmed by simulation tests using a computer simulation program of a roller press. The simulator was developed by the employees of the Department of Manufacturing Systems of AGH University of Science and Technology, based on the Hryniewicz mathematical model of briquetting in a roller press [23]. The model used the thin layer method. This method consisted of the separation in the compaction zone, the volume elements of the briquetted material limited by the side surfaces of the rollers, their sealings, and two planes perpendicular to the direction of the material movement, distant from each other by an infinitely small *dy* value. To determine the relationship between unit forces and stresses on the surface of a separated element, the equilibrium condition of the forces acting on it was used. The program determined the maximal unitary pressure value exerted on the briquettes in the central zone of the molding cavities, as well as the loads caused in a drive and hydraulic system [45]. When performing the briquetting simulation test on the material, it was necessary to determine the material compaction pressure characteristic (ϑ) and the variability of friction static coefficient (µ_s_). In this case, the test of material compaction pressure characteristic (ϑ) and the variability of friction static coefficient (µ_s_) were performed, using cooper ore concentrate with a 11% sulphite lye binder and were obtained by Equations (4) and (5) as follows:ϑ = 0.13165 s ^14.68478^ w ^−1.55615^(4)
where ϑ represents the material compaction pressure characteristic, s is the compaction level, and w is the moisture content.
µ_s_ = −1.35629s − 0.08215w + 3.01968(5)
where µ_s_ is the coefficient of static friction, s is the compaction level, and w is the moisture content.

The 7.0° grip angle and mixture moisture content of 4.2% were used in the simulation. It was assumed that there was no wear of the rollers. The tests also showed that under extremely unfavorable operating conditions, i.e., the material being overdried to a moisture level of 2% and with the gap minimized to 1 mm, it could have sporadically happened that the pressures exerted on the molding cavity bottom reached over 100 MPa. The simulation process results are presented in Figure 4.

The surface was assessed based on photographs taken by means of a Nikon D5000 camera with the Nikkor 18-105 VR lens (Nikon, Tokyo, Japan), including a macro lens with a 10× zoom for each sample in three places. A scheme of the samples and observation spots is presented in Figure 5.

The 3D surface scanning (Figure 6a) was done by means of a scanner by ROMER (Hexagon MI, Cobham, Surrey, UK). In order to obtain the CAD format, the gathered data were processed in the Geomagic Design X 64 software (3D Systems, Rock Hill, SC, USA) and exported to the SolidWorks 2016 software (Dassault Systèmes S.A., Vélizy-Villacoublay, France). The obtained 3D model (Figure 6b) made it possible to generate virtual cross-sections of the matrix, and therefore to select spots from which samples were taken for further tests.

In order to prevent temperature changes from having an effect on the test results, the matrix was cut with an abrasive waterjet. Areas marked in Figure 7 were selected for metallographic tests. Adequately sampled pieces were intended for further metallographic tests and to be observed by means of the scanning electron microscopy.

Coordinate measurements were performed by means of a Global machine with a corundum head. Metallographic micro sections from samples previously cut out in marked places were replaced with resin inclusions, and then ground and polished. The micro sections were etched with 2% nital. The microstructure was documented using a Carl Zeiss Axiovert 200 MAT microscope (Carl Zeiss Microscopy Deutschland GmbH, Oberkochen, Germany). The microstructure can be observed (Figure 8) on threshold cross-sections at point X marked on the scheme (Figure 5). In Figure 5, the microstructure in the near-surface layer of the samples was observed in three places, marked on scheme A, B, and on the edge AB. The tests were conducted using a scanning electron microscope (Japan Electron Optics Laboratory Co., Ltd., Tokyo, Japan) and consisted of observing abrasion surfaces on thresholds of cavities of the working rings. The hardness measurement of the roller press ring was performed by the Vickers method on a cross-section of a sample taken from the first row, due to its lowest wear degree, using a Zwick/Roell ZHU 187,5 hardness tester (ZwickRoell GmbH & Co. KG, Ulm, Germany). The first measurement was done at a distance of 0.5 mm away from the face and consecutive measurements were along the section presented in Figure 5. The measurement points were 1 mm apart from one another.

The tests showed that the structure was built of tempered martensite or bainite (Figure 8). According to the photographs taken with a graduation scale of 20 µm, the average prior austenite grain size determined by the Bruch surface method was 25 µm, while its average cross-section surface area was 586 µm^2^.

## 3. Results and Discussion

### 3.1. Macro Assessment of the Surface

The observations covered Rows 1, 3, and 5 from the outermost edge of the ring. The first row of the ring suffered the lowest wear degree, because the lowest intensity of the material was fed into this area. One can observe slight abrasions on the thresholds, indentations (area C marked in Figure 9), and scratching. Plastic deformations of thresholds, caused by exceeding the yield strength, are also visible (area E in Figure 9). One can clearly see abrasion on the side walls of the cavities (areas A and B in Figure 9, and area D in Figure 9) and the initial formation of the so-called overhangs which also result from the yield strength being exceeded (area C in Figure 9). In the next analyzed row, the wear process is even more visible. One can clearly see that thresholds are more intensely abraded than in the case of the first row (area J in Figure 9). As thresholds are more and more abraded, their thickness decreases. Plastic deformations are clearly visible in the area connecting thresholds and nodes (areas F and G in Figure 9). More and more overhangs occur which makes it more difficult for the briquettes to leave cavities (areas H and I in Figure 9). The rated geometrical dimensions of the molding cavity change more and more. The wear processes in the fifth row are most intense. The geometry of the cavity clearly is not able to ensure a proper quality of the briquette. Plastic deformation is clearly observable on the indentation surface (areas K and N in Figure 9) and one can see numerous indentations (area M in Figure 9). One can also see deep scratching in the peripheral direction. The remains of the consolidated material are visible on the surface of the scratching, which has an immediate effect on the consolidation environment. Indentations occur which might have been caused by hard particles penetrating the compaction area. The progressing process of the threshold wear has consequentially led to its chipping (area L in Figure 9).

The macro assessment of the surface reveals the complexity of the wear mechanisms, which include abrasion, scratching, chipping, and plastic deformation. Moreover, the observed mechanism does not differ in relation to the few technical and scientific reports in this area [26,30]. The lack of publications in this area is understandable in the context of protecting the results of many years of research and experience related to the design, construction, and operation of roll presses also equipped with molding rings.

### 3.2. Aerological Macro Observations

The surface topography of thresholds was documented on both sides of the mold (markings A and B in Figure 5) and on an edge which was formed in the process of wear (marking AB in Figure 5). It is presented in Figure 10.

The surfaces of all samples showed traces of corrosive processes and abrasive wear. For side A (Figure 10a–c), the intensity of occurrence of scratches, abrasions, and corrosive spots increases with the consecutive row numbers. Considering the observations of surface B (Figure 10g–i), the intensity of scratches and indentations is highest for the fifth-row sample. The black areas are probably remains of the briquetted material. On edge AB (Figure 9d–f), one can notice indentations and chippings. Here, the erosive process is more visible due to the thinning of the tool wall. One can also see numerous scratches and plastic deformations of the edge, caused by the consolidated material grains being dented into the ring material, and then by its motion. Numerous spalling and pitting wear areas are visible. The observed sharpening in the AB area is an expected phenomenon due to the nature of the tool’s operation and is confirmed in the literature [26]. This process is associated with the outflow of material from the merging zone in the final phase of agglomeration, i.e., when the mating molding rings of both rolls are open and the already integrated material leaves the molding cavity.

### 3.3. Geometry of Wear

The obtained point coordinates are graphically presented in Figure 11.

In order to determine losses of material of the molding cavities, an output contour was applied onto the contours obtained (Figure 12). In order to determine the wear, it was assumed that the measurement would be performed for one-quarter of the cavity size (angle 2.09° in relation to the working ring threshold axis).

It was noticed in the result of the analysis that the lower part of the molding cavity suffered greater material loss. On the basis of the obtained drawings, the wear surface area was determined. The SigmaScan Pro software (Systat Software Inc., San Jose, CA, USA) was used to measure the areas. The results are presented in Table 2.

The observed geometry of wear differs from the one suggested in [30,31]. A clear threshold occurs in the tip area. This may suggest that the resistance to wear decreases towards the depth of the surface, which may be an effect of the surface treatment. The results obtained confirm that the wear of a specific row is related to its distance from the feed walls and Row 1 demonstrates the lowest material loss. The wear increases towards the axis of the roller press ring. Considering the left side (A) and right side (B) of the threshold, the wear in the molding cavity is asymmetrical. In the case of the first and third row, side A suffered greater wear than side B. The inverse occurred in the fifth row, which probably resulted from significantly greater deformation.

### 3.4. Microstructure

For the first row of the sample, the microstructure was documented on side A along the straight line starting at a distance of 1.5 mm from the edge with 1 mm spacing (Figure 13). According to the microstructure changes observed, differences occur as the distance from the working surface increases. For a distance up to 1.5 mm, it is characterized with noticeably finer grain size with a coniferous-like martensite structure. Microstructures observed at distances of 3.5–8.5 mm are similar. They are characterized with the microstructure having morphology corresponding to the upper bainite.

Figure 14 and Figure 15 show an exemplary documentation of the microstructure in the near-surface layer, for areas A, AB, and B. The microstructure confirms carburization of the surface layer to a depth of about 2 mm.

As a result of the near-surface layer analysis of the sample taken from the first row, initiated wear processes can be seen. In the near-surface area for edge A, one can notice micro-chippings and micro-cracking (Figure 14). In the threshold tip area (AB), its apex plastic deformation effect is visible. The microstructure in this area is fine grained. Clearly, the further away from the edge, the stronger the granularity becomes. In the area of transition from area AB to B, the edges become rounded, which is related to the wear process. The carburized layer with a coniferous-like martensite structure is clearly visible. While looking at the edge of the working surface cross-section of the sample taken from the third row, one can see numerous areas in which chipping and cracking have occurred. One can see numerous places where material has been torn out. The documentation made for side A shows that it has been more intensely worn as compared with side B. The near-surface areas which are brighter and etched may suggest that the so-called white etching layer is being formed. While analyzing the surface layer microstructure in the threshold area for the fifth row (Figure 15), one can spot changes in the mechanism of wear depending on the position against the mentioned above threshold. The microstructure documentation made in areas A and AB is similar, indicating abrasive wear as compared with area B. Fatigue wear effects and numerous chippings are visible in area B. This could be related to the fact that the wear process had occurred within a certain range, and then reached such a point that the threshold was torn off in area B. Similar to the third row of the sample, the prevailing wear mechanism consisted of chipping and cracking. In this case, an area with morphology corresponding to the so-called white etching layer was observed. This might indicate an abrasive effect taking place in this micro area, resulting from the heating up to the austenitizing temperature, and then intensive cooling [22]. In the case of samples from the third and fifth row, no carburized layer was observed and it was mostly worn out. Presumably, in the initial stage of operation, the carburized layer delayed the wear process; however, when it was abraded off, its wear became much more intense.

### 3.5. Surface Observation with SEM

Surface observations were made using an electron microscope, similar to the observations made with a macro lens according to the diagram presented in Figure 5. Figure 16a–c presents summarized topographic documentation of the surface observed in area A for the first, third, and fifth rows. In Figure 16a, one can observe craters, the initiation of which can be related to the corrosive wear, and then the fatigue wear resulting in the chipping of the material. Their interior is characterized with high roughness, which intensifies further growth of the material loss in this place. Scratches are also visible. Chipping, ranging from 30 to 50 µm, can be observed. In the case of the third row, chipping areas are less intense. The effects of abrasive wear begin to be visible. In Figure 16b, one can clearly see areas with a bright and dark shade. Areas situated higher are brighter. The existence of such areas can be related to plastic deformation (material flow plasticity). This phenomenon originates from the difference in the degree of deformational reinforcement between micro areas which, as a result of that, show different abrasive resistance. According to the surface topography obtained from the fifth row (Figure 16c), one can also observe chipping and cavities formed under the impact of the consolidated material particles. However, the prevailing wear mechanism is the abrasive wear, which is proven by numerous scratches and areas of accumulation of the abraded material. Edge AB is the area where the wear mechanisms are most intense. In the first row, numerous cavities left by the chipped material are seen. It is also possible to observe the fatigue wear mechanism, that is, spalling. It consists of pieces of material that tear off in the form of flakes (spalling). In Figure 16e (third row), material plastic deformations have been observed which may be related to the process of material particles material being consolidated by micro cutting and chasing the surface of the molding cavities. Areas, where the material was produced as a result of the abrasive wear on the outermost part of the edge, can also be seen. In the fifth row (Figure 16f), the threshold has been entirely destroyed, and therefore the surface is characterized with morphology corresponding to abrasive wear. Side B in the first row is strongly scratched (Figure 16g). The scratches are directional, which is related to the direction of rotation of the working rings (movement of the briquetted material against the working rollers). The spalling process effects are most visible in Figure 16h (third row). The lower areas where material has been torn off are clearly distinguishable. The abrasive wear processes are also intense (numerous scratches caused by micro cutting and chasing). The surface topography on side B of the fifth row shows morphology corresponding to weak abrasive wear.

These observations confirm the previously obtained results in terms of the complexity of the wear mechanisms, which include abrasion, scratching, chipping, and plastic deformation.

### 3.6. Hardness Measurements

Hardness measurement results are presented in Figure 17. Changes of hardness on the cross-section indicate a heat and chemical treatment (technologically, carburization) that the ring underwent at the stage of its production. The highest hardness was obtained at the threshold face, within the carburized layer area. As the distance from the face increases, it visibly decreases to a depth of approximately 2 mm, where it reaches a value of approximately 260 HV. Hardness becomes stable in this area. Evaluation of the general tendency of hardness changes on the cross-section of samples taken from the first row, i.e., the area characterized by the lowest wear, included a hardness of steel at the level of 100–150 HV and a hardness close to 350 HV obtained at a depth of 0.5 mm, therefore it is difficult to expect the required hardness of 700 HV on the surface. Furthermore, it is not difficult to achieve the required level of hardness for 20Cr4 steel after proper treatment [46,47]. Carburizing and quenching in a polymer solution, at a temperature of 30 °C, obtains a harness of 954 HV [47]. On the basis of the results obtained, we suggest a way to control the wear of the working rings. If the working ring measurement produces a wear result at a level of 3 mm, it should be assumed that the wear process will continue with higher intensity. This effect should be taken into account in the ring operation time frame.

## 4. Conclusions

On the basis of the test results obtained, it is possible to determine the types and main mechanisms of wear on working rings in roller presses.

The working rings are non-uniformly worn. The smallest wear affects the molding cavities situated on the outermost edges of the ring. The wear increases as the center of the ring is approximated, and it reaches its maximum at mid-width. This results from the way in which the consolidated material is fed in the compaction zone.The molding cavities are worn asymmetrically. This is related to the direction in which the roller rings rotate. In the case of the shape considered in the study, the lower part of the cavity is subjected to a higher wear rate.The working rings are subjected to a carburization process which results in an increase in the surface layer hardness to approximately 370 HV. The required hardness should amount to 700 HV; it should be remembered, however, that the surface is abraded. While analyzing the carburized layer morphology, it can be concluded that, once a wear rate of about 3 mm has been reached, the wear continues with much greater intensity. The roller ring microstructure changes depending on the distance from the cavity face, which is related to the process of carburization and further heat treatment.As the wear becomes more advanced, its mechanisms change. At the initial stage, this is mainly chipping and cracking, started by corrosion processes, which then turn into spalling processes, and finally into abrasive wear.The top of the threshold undergoes the characteristic wear. This is related to high unit pressures in this area which often exceed the yield strength and to the previous deterioration and removal of the carburized layer by the loss of material.

The wear analysis presented herein may be used for a conscious choice of geometrical parameters and materials for producing roller press working rings in order to extend their operational lifetime and the kinetics of wear.

## Figures and Tables

**Figure 1 materials-13-05782-f001:**
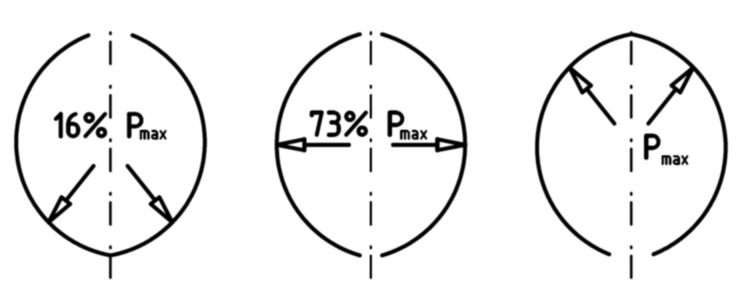
Unit pressure value distribution during the briquetting process in a roller press. P_max_, maximum unit pressure [23].

**Figure 2 materials-13-05782-f002:**
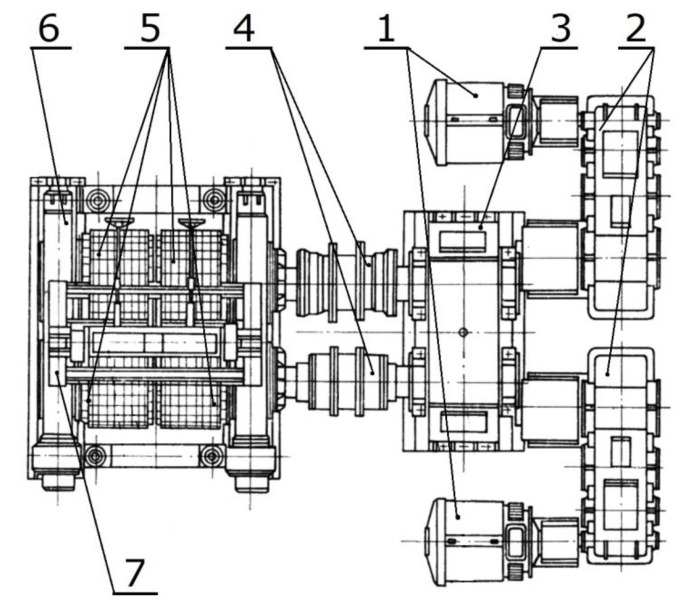
Scheme of roller press where the molding ring used for tests worked. (**1**) Electric drive; (**2**) Gearbox; (**3**) Synchronizing gearbox; (**4**) Clutch; (**5**) Molding rollers; (**6**) Frame with hydraulic system; (**7**) Feeder [43].

**Figure 3 materials-13-05782-f003:**
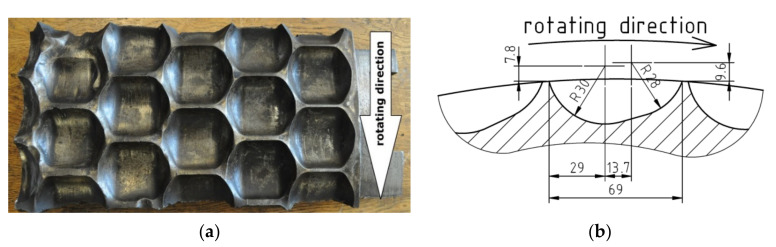
Molding ring. (**a**) Piece used for tests; (**b**) Molding cavity geometry.

**Figure 4 materials-13-05782-f004:**
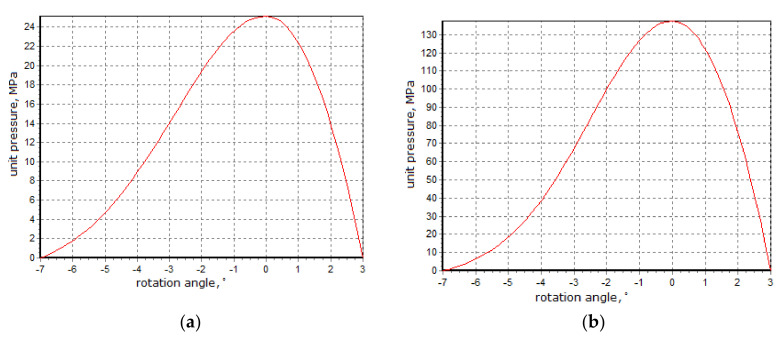
Results of the simulation of the pressure exerted on the bottom of the molding cavity in a press with rollers of a diameter of 1000 mm while briquetting copper concentrates with sulphite lye. (**a**) Regular conditions; (**b**) Extremely unfavorable conditions.

**Figure 5 materials-13-05782-f005:**
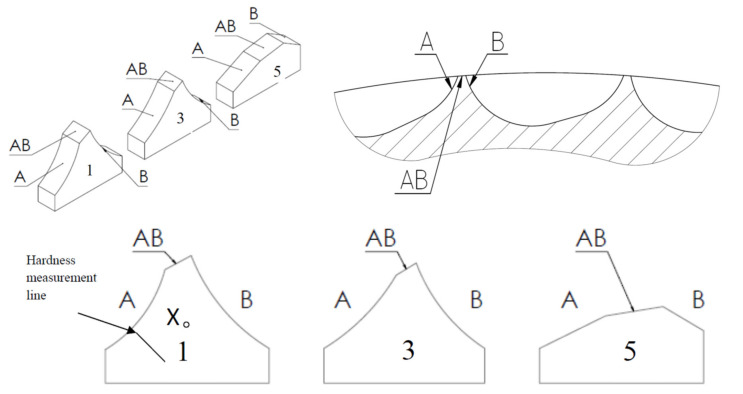
Scheme of samples taken from the first, third, and fifth row with a division per observation zones. (**A**) The lower side of mold; (**B**) The upper side of mold; (**AB**) The thresholds; (**X**) The point of microstructure observation.

**Figure 6 materials-13-05782-f006:**
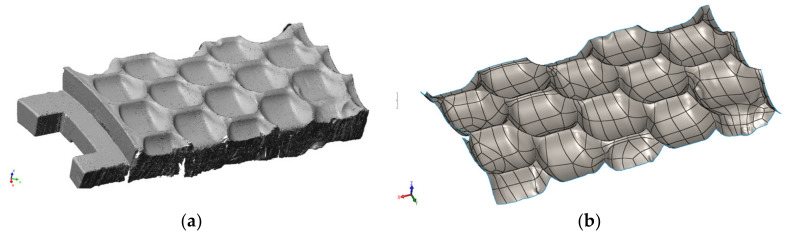
Matrix three-dimensional (3D) model. (**a**) Result of the matrix optical scanning in a form of a cloud of points, done by means of the ROMER measuring arm; (**b**) Obtained in the SolidWorks 2016.

**Figure 7 materials-13-05782-f007:**
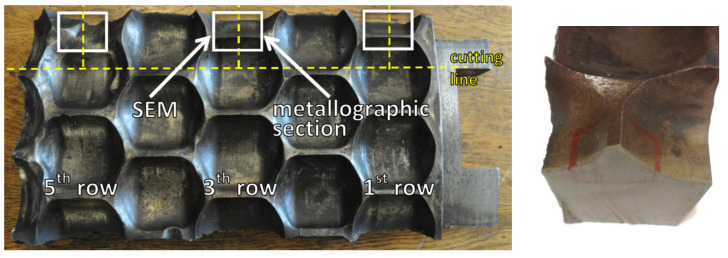
Place of sampling for further tests.

**Figure 8 materials-13-05782-f008:**
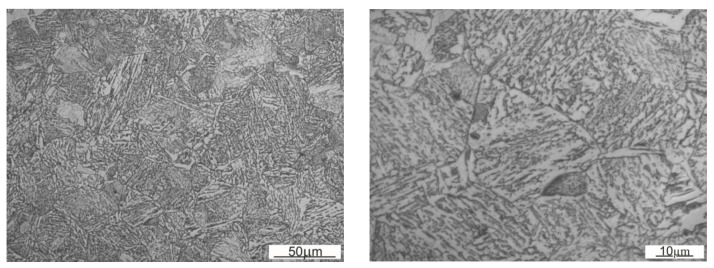
Microstructure of roller press molding ring made of 20Cr4 steel.

**Figure 9 materials-13-05782-f009:**
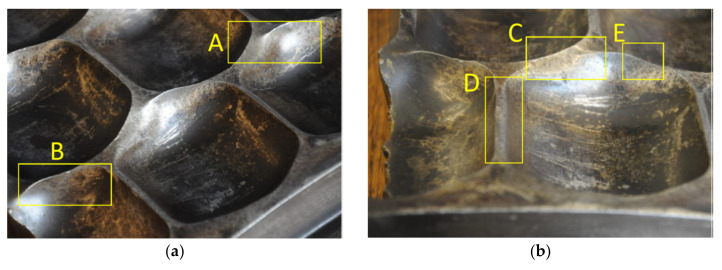
Ring surface topography. (**a**,**b**) First row; (**c**,**d**) Third row; (**e**,**f**) Fifth row.

**Figure 10 materials-13-05782-f010:**
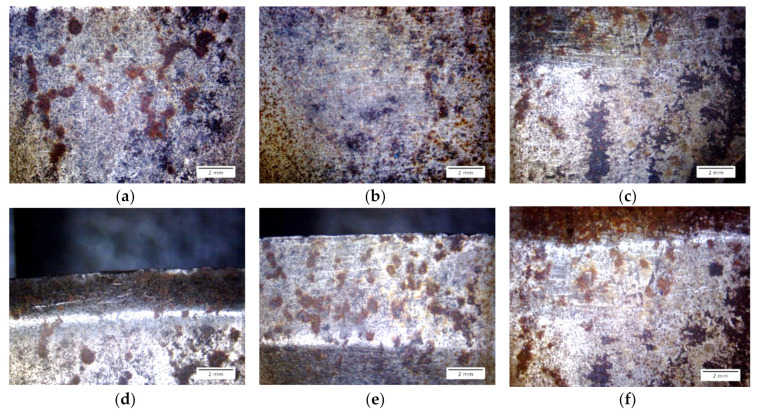
Observation of threshold abrasion surfaces with a macro lens. On side A of the samples. (**a**) First row; (**b**) Third row; (**c**) Fifth row. On side AB of the samples. (**d**) First row; (**e**) Third row; (**f**) Fifth row. On side B of samples. (**g**) First row; (**h**) Third row; (**i**) Fifth row.

**Figure 11 materials-13-05782-f011:**
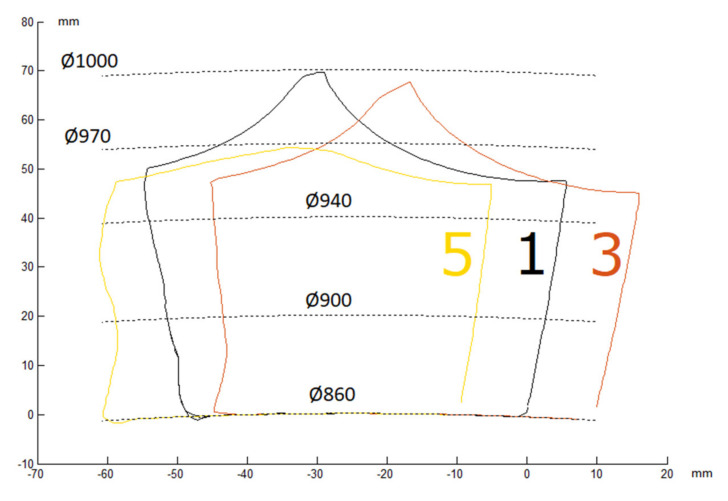
Summary of geometries of the first row, third row, and fifth row cross-sections.

**Figure 12 materials-13-05782-f012:**
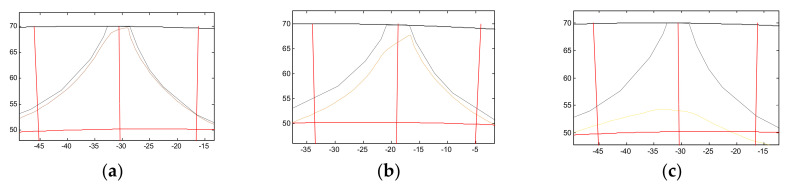
Sections with the superimposed initial profile. (**a**) Row 1; (**b**) Row 3; (**c**) Row 5.

**Figure 13 materials-13-05782-f013:**
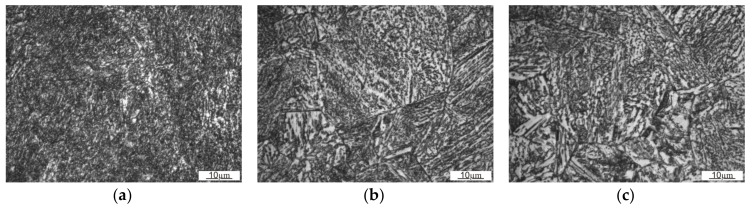
Change of the microstructure in a line perpendicular to surface A, row one, at different distances. (**a**) 1.5 mm; (**b**) 4.5 mm (**c**) 8.5 mm.

**Figure 14 materials-13-05782-f014:**
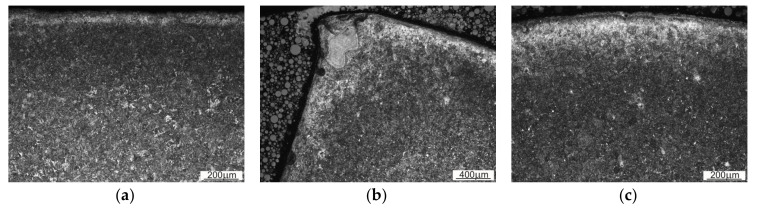
Microstructure within the edge area of the first row of the sample. (**a**) Side A; (**b**) Edge AB; (**c**) Point of transition between edge AB and B.

**Figure 15 materials-13-05782-f015:**
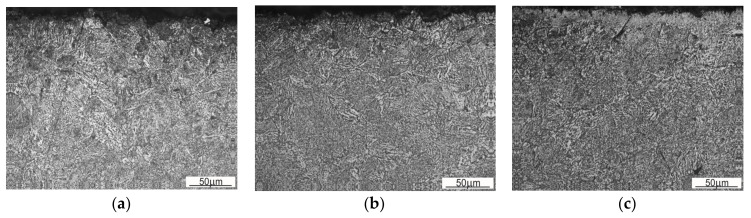
Microstructure within the edge area of the sample from the fifth row. (**a**) Side A; (**b**) Edge AB; (**c**) Side B.

**Figure 16 materials-13-05782-f016:**
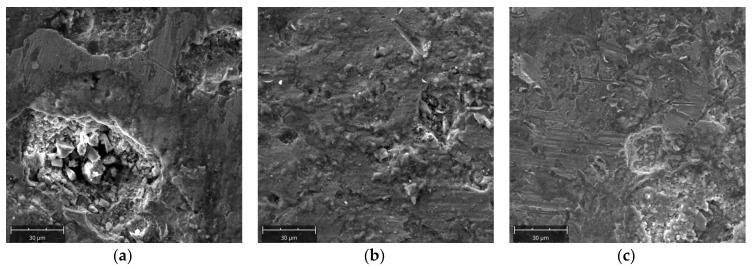
Topography of the abrasion surfaces of molding cavities. Observed on side A. (**a**) First row; (**b**) Third row; (**c**) Fifth row. Observed on side AB. (**d**) First row; (**e**) Third row; (**f**) Fifth row. Observed on side B. (**g**) First row; (**h**) Third row; (**i**) Fifth row.

**Figure 17 materials-13-05782-f017:**
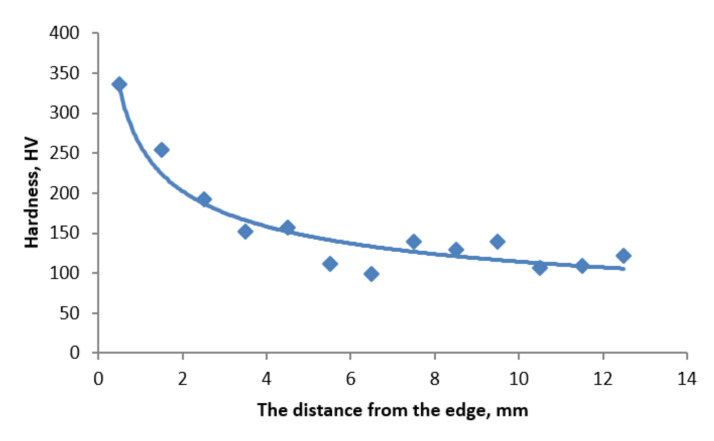
Change of hardness at a distance from the working surface.

**Table 1 materials-13-05782-t001:** Molding ring chemical composition (optical emission spectroscopy).

Chemical Composition, Mass %											
C	Si	Mn	S	P	Cr	Ni	Ti	Co	Al	Mo	Fe
0.22	0.29	0.83	0.0085	0.016	1.04	0.088	0.063	0.008	0.047	0.022	rest

**Table 2 materials-13-05782-t002:** Loss of material on the threshold cross-section of the first, third, and fifth rows.

	Side A		Side B	
	Loss, %	Loss, mm^2^	Loss, %	Loss, mm^2^
Row 1	8	13	6	9
Row 2	30	50	14	21
Row 3	72	120	86	130

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
