# Peer review of "The Wear on Roller Press Rollers Made of 20Cr4/1.7027 Steel under Conditions of Copper Concentrate Briquetting"

_materials, 2020, doi:10.3390/ma13245782_

Round 1

Reviewer 1 Report

In this manuscript, a wide ranging analysis of the wear of a carbon steel that has been applied to the briquetting of copper ore concentrate. The experimental work was performed in-situ in a production environment to replicate the actual conditions seen in service. The experimental specimen was examined in a variety of ways, including visual, profile measurement, optical microscopy, SEM and hardness.

While the use of actual work pieces in this experimental captured the corresponding operating conditions, this is not beneficial to the repeatability of the test. In this study, only one specimen was analysed. Can the authors comment on the likely variability in the operating conditions (aside from the moisture content) and thus their likely affect on the wear?

The authors presented simulated pressures resulting from the process itself (Fig 4). There are insufficient details to enable this simulation results to be assessed. For example, no information was presented as to how the simulation was performed. For example, what mesh was the finite element analysis performed on, what are the material models and properties that were applied? In addition, substantial amount of wear occurred in the sample which would substantially affect the pressure profile. Was this reflected in the simulation?

The micrographs in fig 10 are difficult to read. In particular, the changes in the “scratches, abrasions and corrosive spots” over the three rows that are key to the results are not obvious.

The authors “suggest that the resistance to wear decreases towards the depth of the surface, which may be an effect of the surface treatment.” However, the authors have not ruled out the change in moulding cavity geometry due to wear that can also potentially cause this accelerated wear behaviour. Can the authors please comment on this aspect?

The authors have attributed the loss of material in Fig 16a to corrosive and fatigue wear while that of Fig 16b is attributed to plastic deformation. Can the authors please clarify how and why this distinction was made?

There are also a number of typographical errors. For example, there is no Fig 9g, “raw” instead of “row” and so on.

Author Response

Thank you very much for taking the time to read our manuscript thoroughly and make recommendations for its correction and improvement. We have read the comments carefully and have referred to all your comments. Changes in the article are marked in yellow.

Remark 1:

Can the authors comment on the likely variability in the operating conditions (aside from the moisture content) and thus their likely affect on the wear?.

Answer 1:

Thank You for this remark. We have expanded the article with the information below:

The wear of the moulding surfaces is related to the deterioration of the mechanical properties of briquettes. The volume of the forming cavities increases with wear of the molding surfaces. Increasing the volume of briquettes results in reducing the unit pressure, which in extreme cases may be insufficient to properly merge the briquette or can cause the material to stay in the worn moulding cavities. This results in an increase of loads in the compaction systems and the drive system of the press as well as a decrease of its output capacity [22,32]. The wear of the molding surface can also lead to an increase in the gap between the rolls, which also results in a reduction in the pressures exerted in the molding cavity. Our own experience shows that the wear of rollers is greatly influenced by the properties of the briquetting material, especially the external friction coefficient, hardness and abrasive properties. The actions aiming at accurate identification of wear mechanisms and also the intensity of these phenomena over time require to be accurately understood and accurately verified on real objects.

Remark 2:

The authors presented simulated pressures resulting from the process itself (Fig 4). There are insufficient details to enable this simulation results to be assessed. For example, no information was presented as to how the simulation was performed. For example, what mesh was the finite element analysis performed on, what are the material models and properties that were applied? In addition, substantial amount of wear occurred in the sample which would substantially affect the pressure profile. Was this reflected in the simulation?

Answer 2:

Thank You for this remark. We have expanded the article with the information below and added a reference where the model and simulation program is described:

This has been also confirmed by simulation tests. The computer simulation program of a roller press was used. The simulator was developed by the employees of the Department of Manufacturing Systems of AGH University of Science and Technology, based on the Hryniewicz mathematical model of briquetting in a roller press [23,43]. The model used the thin layer method. This method consisted, in the separation in the compaction zone, of the volume elements of the briquetted material limited by the side surfaces of the rollers, their sealings, and two planes perpendicular to the direction of the material movement, distant from each other by an infinitely small dy value. To determine the relationship between unit forces and stresses on the surface of a separated element, the equilibrium condition of the forces acting on it was used. The program allowed determination of the maximal unitary pressure value exerted on the briquettes in the central zone of the molding cavities, also loads caused in a drive and hydraulic system. When performing a simulation test of the briquetting process of a material, it was necessary to determine the material compaction pressure characteristic (J) and the variability of friction static coefficient (ms). In this case, the test of material compaction pressure characteristic (J) and the variability of friction static coefficient (ms) were performed, using the cooper ore concentrate with a 11% sulphite lye binder and they take in the form (4) and (5).

J = 0.13165 s 14,68478 w -1,55615

(4)

where:

J is the compaction pressure characteristic,

s is the compaction level, and

w is the moisture content.

ms = –1.35629s – 0.08215w + 3.01968

(5)

where:

ms is the coefficient of static friction,

s is the compaction level, and

w is the moisture content.

The 7.0° grip angle and mixture moisture content of 4.2% were used in the simulation. It was assumed no wear of the rollers. These tests have also shown that under extremely unfavourable operating conditions, i.e. the material being overdried to a moisture level of 2% or with the gap minimised to 1 mm, it could have sporadically happened that the pressures exerted on the moulding cavity bottom reached over 100 MPa. The simulation process results are presented on Figure 4.

  1. Bembenek, M. Exploring Efficiencies: Examining the Possibility of Decreasing the Size of the Briquettes Used as the Batch in the Electric Arc Furnace Dust Processing Line. Sustainability 2020, 12, 6393.

Remark 3:

The micrographs in fig 10 are difficult to read. In particular, the changes in the “scratches, abrasions and corrosive spots” over the three rows that are key to the results are not obvious.

Answer 3:

The micrographs in Fig. 10 are slightly brighter. Unfortunately, better showing the surface is difficult - the image does not reflect the view of the surface when viewed stereoscopically.

Remark 4:

The authors “suggest that the resistance to wear decreases towards the depth of the surface, which may be an effect of the surface treatment.” However, the authors have not ruled out the change in moulding cavity geometry due to wear that can also potentially cause this accelerated wear behaviour. Can the authors please comment on this aspect?

Answer 4:

The mold cavity geometry adopts a shape optimized under various conditions. Although the surface was carburized, the resulted of work is a change in the geometry of the seat. Consequently, due to the change of the geometry to a less favorable one, the wear could be accelerated.

Remark 5:

The authors have attributed the loss of material in Fig 16a to corrosive and fatigue wear while that of Fig 16b is attributed to plastic deformation. Can the authors please clarify how and why this distinction was made?

Answer 5:

The observe corrosive craters are marked on Fig 16a. The presumptive directions of furrowing are marked on Fig 16b with thick arrows and the plastic flow of the material to the sides (thin arrows).

Remark 6:

There are also a number of typographical errors. For example, there is no Fig 9g, “raw” instead of “row” and so on

Answer 6:

Indeed, it could be confusing – there are Fig 9e and 9f, “raw” was changed to “row”.

Once again, thank you very much for the comments, the consideration of which helped us improve the manuscript.

Reviewer 2 Report

This manuscript reported an experimental investigation on the wear problem of the roller press rollers by 20Cr4 steel under the working condition of copper concentrate briquetting. The results of the study will be of good reference value to the improvement of the production process of copper concentrate briquetting. However, I believe that the issue studied in this paper is relatively too specific, it is difficult to arouse the general interest in the material research community. Therefore, I don't think the manuscript is suitable for publication in the journal "Materials". It is suggested that the authors should try resubmit their manuscript to “Journal of Manufacturing and Materials Processing” for the possible publication.

Author Response

Dear Reviewer,

Thank you very much for taking the time to read our manuscript thoroughly and make recommendations.

Reviewer 3 Report

Dear authors,

Thank you for your article investigating an important topic in metal processing industry. Below you will find my comments.

General comments:

Overall the English is quite poor, the sentences are sometimes hard to understand and there are many grammatical mistakes. The article would greatly benefit from being submitted to a English editing service. As they are to numerous to mention I won't include to many in my subsequent comments.

Nice microstructure images

Abstract:

  • What do you mean by macroscopy? Is this visual inspection or mistype and you mean optical microscopy?
  • Should include more of your conclusions

Introduction:

  • The final paragraph is more like a first introduction paragraph of why this is done.
  • Are there any similar studies investigating wear of roller press rollers, maybe for other materials?
  • Add a final paragraph on what you will be doing in this study. What are you investigating and how. 

Materials and Methods:

  • What is the unit Mg in line 122?
  • Can yo add labels to Fig 2 to make it easier to read?
  • Please stick to one set of units. HRC or HV
  • Add reference to software used.
  • Line 189, this preparation was for both the optical microscope and SEM? Please clarify
  • Fig 8a and Fig 14, can be removed, not much useful information at this scale.

Results and Discussion:

  • Little to no discussion in 3.1, 3.2, 3.5 and 3.6.
  • The whole section is more like a results section. Needs a more discussion of the results.

Conclusion:

  • More like a discussion and summary than conclusion. 
  • Make the conclusion more clear, perhaps a bullet list?

Author Response

Dear Reviewer,

Thank you very much for reviewing our manuscript. According to the comments and the questions, we have carefully revised the article text. Below we answer to all the remarks. The changes in the manuscript have been highlighted by yellow colour.

Remark 1:

Nice microstructure images.

Answer 1:

Thank you very much for your support. This is very motivating for us.

Abstract:

Remark 2:

What do you mean by macroscopy? Is this visual inspection or mistype and you mean optical microscopy? Should include more of your conclusions

Answer 2:

By macroscopy we mean visual examinations using a microscope with magnifications not exceeding 50x. The abstract was extended with selected conclusions (keeping the maximum number of words in this section).

Introduction:

Remark 3:

The final paragraph is more like a first introduction paragraph of why this is done.

Answer 3:

Indeed, it could be confusing, but we consider that the final paragraph of the introduction is a continuity of the information given in the introduction and leads to the presentation of the detailed topic of the manuscript. Last paragraph of the manuscript (conlusions) has been rewrite as noted in the Remark 13.

Remark 4:

Are there any similar studies investigating wear of roller press rollers, maybe for other materials?

Answer 4:

Unfortunately, the problem of wear of the forming rollers of briquetting roller press is not widely described in the world literature. The items available and known to the authors have been included in the references [26,30,32,33]. On the other hand, it opens up new possibilities and research fields.

Remark 5:

Add a final paragraph on what you will be doing in this study. What are you investigating and how.

Answer 5:

Indeed, it is not sufficiently explicitly described. We have added basic information to the introduction. The detail of the research are available in the section materials and Methods.

Materials and Methods:

Remark 6:

What is the unit Mg in line 122?

Answer 6:

A Mg (megagram) is derived from the SI unit of 1,000 kilograms

Remark 7:

Can you add labels to Fig 2 to make it easier to read?

Answer 7:

Thank You for this remark. We added labels to Fig 2.

1— electric drive, 2—gearbox, 3—synchronizing gearbox, 4—clutch, 5—moulding rollers, 6—frame with hydraulic system, 7— feeder

Remark 8:

Please stick to one set of units. HRC or HV.

Answer 8:

Thank You for this remark. We unified units.

Remark 9:

Add reference to software used.

Answer 9:

Thank You for this remark. We added references to the sofrware.

Remark 10:

Line 189, this preparation was for both the optical microscope and SEM? Please clarify.

Answer 10:

The microstructure after etching was observed on the cross-section of the samples using a metallographic microscope. The surface observations were made by SEM.

Remark 11:

Fig 8a and Fig 14, can be removed, not much useful information at this scale.

Answer 11:

Fig 8a has been removed. We consider that Fig. 14 should be left as evidence of a carburized layer.

Results and Discussion:

Remark 12:

Little to no discussion in 3.1, 3.2, 3.5 and 3.6.

The whole section is more like a results section. Needs a more discussion of the results.

Answer 12:

Thank you for this suggestion. Indeed, these chapters focused mainly on the description of the observations made because it was difficult to discuss these observations only in relation to individual chapters. However, they were necessary to formulate the main observations discussion and conclusions. Nevertheless, as suggested, we tried to broaden the scope of the discussions in the individual chapters.

Conclusion:

Remark 13:

More like a discussion and summary than conclusion. Make the conclusion more clear, perhaps a bullet list?

Answer 13:

Thank You for this remark. We re-edited this section.

Thank you very much for your constructive comments. We are convinced that the amendments have significantly improved the quality of our work.

Round 2

Reviewer 1 Report

This revision addresses the points raised satisfactorily. 

Reviewer 2 Report

The manuscript is the revised version. The authors answered all the questions raised by the reviewers appropriately, and modified the contents of the manuscript accordingly. The research results reported in this paper are detailed and comprehensive, which have reference value for other people's further research. Therefore, I recommend that the manuscript be published in the Materials journal.

Reviewer 3 Report

Thank you for your improvements. The manuscript is better and easier to follow.